# Migration of Small Ribosomal Subunits on the 5′ Untranslated Regions of Capped Messenger RNA

**DOI:** 10.3390/ijms20184464

**Published:** 2019-09-10

**Authors:** Nikolay E. Shirokikh, Yulia S. Dutikova, Maria A. Staroverova, Ross D. Hannan, Thomas Preiss

**Affiliations:** 1EMBL-Australia Collaborating Group, Department of Genome Sciences, The John Curtin School of Medical Research, The Australian National University, Canberra, ACT 2601, Australia; 2State University of Social and Humanitarian Studies, Kolomna, Moscow Region 140410, Russia; 3Institute of Protein Research, Russian Academy of Sciences, Pushchino, Moscow Region 142290, Russia; 4Australian Cancer Research Foundation Department of Cancer Biology and Therapeutics, The John Curtin School of Medical Research, The Australian National University, Canberra, ACT 2601, Australia; 5Sir Peter MacCallum Department of Oncology, University of Melbourne, Parkville, VIC 3010, Australia; 6Department of Biochemistry and Molecular Biology, Monash University, Clayton, VIC 3800, Australia; 7Department of Biochemistry and Molecular Biology, University of Melbourne, Parkville, VIC 3010, Australia; 8School of Biomedical Sciences, University of Queensland, Brisbane, QLD 4072, Australia; 9Victor Chang Cardiac Research Institute, Darlinghurst, NSW 2010, Australia

**Keywords:** eukaryotes, gene expression control, mRNA translation, translation initiation, ribosomal scanning, SSU, 40S ribosomal subunit, cap-dependent initiation, eIF4A, eIF4F, 5′UTR, 5′ UTR

## Abstract

Several control mechanisms of eukaryotic gene expression target the initiation step of mRNA translation. The canonical translation initiation pathway begins with cap-dependent attachment of the small ribosomal subunit (SSU) to the messenger ribonucleic acid (mRNA) followed by an energy-dependent, sequential ‘scanning’ of the 5′ untranslated regions (UTRs). Scanning through the 5′UTR requires the adenosine triphosphate (ATP)-dependent RNA helicase eukaryotic initiation factor (eIF) 4A and its efficiency contributes to the specific rate of protein synthesis. Thus, understanding the molecular details of the scanning mechanism remains a priority task for the field. Here, we studied the effects of inhibiting ATP-dependent translation and eIF4A in cell-free translation and reconstituted initiation reactions programmed with capped mRNAs featuring different 5′UTRs. An aptamer that blocks eIF4A in an inactive state away from mRNA inhibited translation of capped mRNA with the moderately structured β-globin sequences in the 5′UTR but not that of an mRNA with a poly(A) sequence as the 5′UTR. By contrast, the nonhydrolysable ATP analogue β,γ-imidoadenosine 5′-triphosphate (AMP-PNP) inhibited translation irrespective of the 5′UTR sequence, suggesting that complexes that contain ATP-binding proteins in their ATP-bound form can obstruct and/or actively block progression of ribosome recruitment and/or scanning on mRNA. Further, using primer extension inhibition to locate SSUs on mRNA (‘toeprinting’), we identify an SSU complex which inhibits primer extension approximately eight nucleotides upstream from the usual toeprinting stop generated by SSUs positioned over the start codon. This ‘−8 nt toeprint’ was seen with mRNA 5′UTRs of different length, sequence and structure potential. Importantly, the ‘−8 nt toeprint’ was strongly stimulated by the presence of the cap on the mRNA, as well as the presence of eIFs 4F, 4A/4B and ATP, implying active scanning. We assembled cell-free translation reactions with capped mRNA featuring an extended 5′UTR and used cycloheximide to arrest elongating ribosomes at the start codon. Impeding scanning through the 5′UTR in this system with elevated magnesium and AMP-PNP (similar to the toeprinting conditions), we visualised assemblies consisting of several SSUs together with one full ribosome by electron microscopy, suggesting direct detection of scanning intermediates. Collectively, our data provide additional biochemical, molecular and physical evidence to underpin the scanning model of translation initiation in eukaryotes.

## 1. Introduction

Ribosomal scanning of mRNA is generally accepted as the predominant mechanism to locate start codons during the initiation phase of eukaryotic translation [1]. The scanning model posits that, upon cap-dependent attachment to mRNA, ribosomal small subunits (SSUs) move in a 5′ to 3′ direction (at the expense of energy; ‘powered’ motion), while the 5′UTR nucleotide sequence is being probed for codon:anticodon and local ‘nucleotide context’ interactions [1,2,3,4,5,6]. Additionally, several cap-independent [7,8,9], or eIF4E (the main cap-binding protein)-independent mechanisms have been proposed over the years to cater for translation under specific conditions and of special mRNAs. These include direct internal initiation by ribosomes, mediated by specialised structural modules such as internal ribosome entry sites (IRESes) found in viral and some cellular mRNAs [10,11,12,13,14], ‘open’ sequences promoting direct or factor-mediated SSU binding [15,16,17] and *N^6^*-methyladenosine modifications [18,19], and 5′-end-dependent but cap-independent initiation [9,20,21], which can be mediated by mRNA:ribosomal (r)RNA basepairing [22] (see more in reviews [23,24,25,26]). Nonetheless, some elements of powered 5′ to 3′ SSU motion remain in place for most of these modified pathways. Notable exceptions are certain viral IRES sequences (of type III or IV) [10,11,12,13] and mRNAs with no [21] or very short 5′-end sequences termed translation initiator of short 5′-UTR (TISU) [27,28], which do not require ‘powered’ scanning. Overall, mRNAs vary greatly in scanning efficiency [29] and it has been shown that scanning factor availability alters transcriptome utilisation [30,31,32], which is a eukaryote-specific feature of gene expression control. Cell responses to external and internal stimuli often target scanning [33,34,35]; the disappearance of the ATP-dependent RNA helicase eIF4A from ribosomal complexes accompanying acute translational shutdown in yeast in response to glucose removal is a prominent example of such responses [36] (reviewed in [37]). Critically, many components of the scanning machinery and particularly the eIF4-group factors are overexpressed in cancers and participate in the onset and maintenance of the malignant phenotype. They are therefore in the focus of active anticancer drug development [33,38,39,40,41]. Thus, insights into the scanning mechanism of translation are valuable from both, the fundamental and medical perspective.

The minimal protein factor requirements for ‘powered’ scanning through moderately structured 5′UTR include eIFs 1, 1A, 2:GTP, 3, 4A, 4B, 4F(4E:4G:4A) and ATP [42,43,44]. More structured 5′UTRs may require additional helicases such as Ded1p in yeast [31], DHX29 in mammals [45], and others [46,47]. However, mRNA with ‘unstructured’ 5′UTRs can assemble SSU complexes at start codons without eIF4-group factors or ATP [16,42,45,48,49,50]. Thus, the mere ‘sliding’ along RNA is an inherent SSU feature [16,51], whereas the directionality of SSU motion needs ‘powered strokes’ [32,52]. eIFs 4F/A/B (and H in mammals) are abundant factors that co-purify with SSUs and collectively have the biochemical ability to separate RNA duplexes while hydrolysing ATP [23,47,48,53,54,55,56,57,58] and are thus considered as the main components of a hypothetical scanning ‘motor’, with eIF4A being the primary candidate for the chemo-mechanical coupling (reviewed in [24,47,55]). It should be noted though, that the RNA strand-separating, ATP-binding and hydrolysis activities of eIF4A change considerably depending on the presence of the other eIF4-group factors [59,60,61,62,63,64], and possibly cap, poly(A) and poly(A)-binding protein, as well as the SSUs themselves [55,65,66]. Regarding scanning, two distinct roles of the eIF4-group factors have been proposed: assisting with initial loading of the SSU onto mRNA and promoting directional motion along the 5′UTR. For the loading phase, it has been shown that ATP-bound eIF4A is required for the displacement of eIF3j and the adoption of an ‘open’, scanning-competent configuration of the SSU [66]. For the directional motion, the major, somewhat diverging propositions are either that eIF4A (assisted by other proteins) ‘clears the way’ in front of the SSUs by removing secondary/tertiary structures [47,56], or that it provides a cycling between high/low affinity stages to the scanning complex, which coupled with other rearrangements, unidirectionally translocates the scanning SSUs along the 5′UTR (a combination of both activities is also possible) ([67]; reviewed in [24]). Direct physical evidence for scanning SSUs is extremely limited, primarily consisting of the original observations that the presence of Edeine (an oligopeptide inhibitor of translation initiation from *Brevibacillus brevis*) in in vitro translation reactions led to polysome-like sedimentation properties of the resultant SSU complexes, which protected ~60 nt regions of mRNA 5′UTRs [2]. More recently, we mapped RNase-resistant ‘footprints’ of scanning SSU complexes by high-throughput sequencing [29]. However, despite much progress, scanning remains a mechanistically enigmatic process among the gene expression control pathways.

In this study, we investigated the consequences of suppressing ATP cycling by eIF4A for translation initiation, using cell-free systems based on mouse Krebs-2 ascites cell lysate or reconstituted from individual mammalian purified components. Unexpectedly, trapping eIF4A in the ATP-bound state inhibited initiation even on an eIF4F/A/B-independent mRNA 5′UTR. In addition to the usual toeprint seen for SSUs at start codons we also detect an unusual SSU toeprint around 8 nt upstream and characterise its factor-dependence and other features. Finally, using electron microscopy we image polysome-like rows of SSUs that assemble in cell-free translation reactions programmed with an mRNA with extended 5′UTR and supplemented with higher concentration of magnesium ions and AMP-PNP, consistent with a proposed cap-severed loading and queuing of multiple SSUs on mRNA [24,29,68,69].

## 2. Results

### 2.1. A Nonhydrolysable ATP Analogue Can Inhibit Translation of Capped Poly(A) 5′UTR mRNA

eIF4A has been shown to be indispensable for scanning of 5′UTRs containing heterogenous sequences and at least some complementarity-based secondary structure [16,42,48,70]. The exact mechanism by which eIF4A acts on mRNA as part of the scanning complexes, however, remains obscure. Treatment with rocaglates (such as Silvestrol and analogues) prevents efficient translation of some mRNAs, likely mediated through reinforcing eIF4A interactions with RNA in a homopurine-dependent manner [71,72]. Rocaglate activity requires neither ATP bound to eIF4A nor the mRNA cap structure. Nevertheless, an analogous inhibitory effect might also be triggered if eIF4A:ATP, rather than providing unwinding activity, was made to bind to RNA persistently enough to prevent progression of the scanning complexes.

We wished to test if these effects of AMP-PNP are dependent on the sequence (and structure) of mRNA 5′UTRs. To this end, we prepared two mRNAs by in vitro transcription, both based on a firefly luciferase (Luc) open reading frame (ORF) and flanked 3′ by sequences from the Tobacco Mosaic Virus 3′UTR. The 5′ flanking sequences were either the β-globin mRNA 5′UTR or a poly(A) stretch (see Figure 5a for the 5′UTR sequences).

Previously, it has been shown that a nonhydrolysable ATP analogue (AMP-PNP) exerts inhibitory effects on translation of mRNA with the β-globin mRNA 5′UTR in cell-free lysates [73]. The β-globin 5′UTR is a classic example of a moderately structured, but eIF4A-, cap- and energy-dependent 5′UTR which requires the complete minimal translation initiation factor set (eIFs 1, 1A, 2, 3, 4A/B/F) and ATP hydrolysis to form initiation SSU complexes over the start codon efficiently [42,44]. In contrast to the β-globin mRNA 5′UTR, the poly(A) 5′UTR has previously been shown to allow eIF4A/B/E/F/G-independent initiation [16,74]. Intriguingly, mRNAs with poly(A) 5′UTRs of variable lengths are characteristic to the postreplicative (‘late’) poxviral mRNAs which encode most of the structural proteins of the virus and are synthesised and translated immediately before the cell lysis [75,76,77,78,79,80,81,82,83,84,85]. A notable feature of the late poxviral mRNAs is that their variable length 5′UTRs are not encoded in the viral genome, but rather emerge as a result of poxvirus RNA polymerase initiation over the ‘late’ promotor sequences comprising nascent RNA strand ‘slippage’ effect [75,76,77,78,79,80]. Poly(A) 5′UTRs of poxviruses were shown to suppress host mRNA translation during the conditions of virus-induced cell stress but concomitantly remain highly translated [86,87,88,89,90]. Previous data have mechanistically explained this observation through the resistance of poly(A) 5′UTR mRNA translation to high concentration of mRNAs in the translation system, with the longer poly(A) 5′UTR (up to 25 nt) resulting in higher translation rates [91]. We have previously demonstrated that in direct competition assays in pure reconstituted system, poly(A) 5′UTR mRNA outcompeted initiation on β-globin 5′UTR mRNA in the presence of the full minimal factor set [16].

Both the β-globin mRNA 5′UTR and poly(A) 5′UTR mRNAs were extensively capped (cotranscriptionally with antireverse m^7^G5′ppp5′G as well as post-transcriptionally; see Materials and Methods for details). We then measured the real-time accumulation of firefly luciferase (Luc) synthesised in a cell-free translation system based on mouse Krebs-2 cell lysate (generally as described in [52]). We assembled the translation system at 0 °C and then programmed it with either of two in vitro synthesised mRNAs. As expected [74], the mRNA with poly(A) 5′UTR does not exhibit ‘self-inhibition’, when present at high concentration, even in its capped form (Figure 1a vs. Figure 1c). This is explained by its relative independence on initiation factors, especially the limiting eIF4F [23,55]. Conversely, mRNA with the β-globin 5′UTR demonstrates strong self-inhibition at 0.45 µM compared to 0.15 µM (Figure 1b vs. Figure 1d), indirectly confirming its high factor dependence in this system. Surprisingly, AMP-PNP addition reduced translation of both mRNAs in a comparable manner. As AMP-PNP does not directly reduce luciferase activity in these conditions (Appendix A), a reasonable explanation is that AMP-PNP strengthens the attachment of some complexes to both mRNAs in a manner that inhibits translation, and potentially scanning. Activity of eIF4A/4F is the most plausible target of such effects, although other ATP/NTP-dependent RNA-helicases might also be responsible for this inhibition.

### 2.2. eIF4A-Blocking RNA Aptamer Does Not Affect Translation of Capped Poly(A) 5′UTR mRNA

To further explore if a specific and competitive inhibition of eIF4A could lead to the same effect as AMP-PNP, we used the high-affinity anti-eIF4A aptamer 20 (a4Aa20; dissociation constant ~40 nM) [92]. a4Aa20 competitively binds to the regular RNA-binding surfaces of eIF4A and thus specifically targets the RNA-helicase of eIF4A, rather than strengthening eIF4A:RNA interactions or affecting its binding to eIF4G [92].

We infused cell-free translation reactions with the in vitro synthesised eIF4A aptamer at the point of reaction assembly, immediately before addition of the mRNAs. We used a4Aa20 at concentrations of 0.08, 0.3 and 1.2 µM (Figure 2), covering the range of effective concentrations established previously [92]. For a control RNA of comparable length, we used *Escherichia coli* 5S ribosomal RNA and scrambled a4AN RNA of the same length as the aptamer (constructed similarly to a4Aa20 RNA; see Materials and Methods) at the same concentrations. The 5S rRNA control did not exert negative effects on either mRNA (Figure 2c,d), confirming specific action of a4Aa20. The anti-eIF4A aptamer had strong mRNA-selective regressive effects on translation (Figure 2a vs. Figure 2b). The total output of translation from the poly(A) 5′UTR mRNA dropped by ~30% at the highest a4Aa20 concentration. This moderate effect can be explained by an earlier exhaustion of the system in these conditions as the rate of product accumulation (i.e., the first derivative of luminescence over time) was not decreased. A similar mild overall translation yield suppression was exhibited by a4AN RNA with β-globin 5′UTR mRNA, possibly due to unspecific initiation factor sequestration (data not shown). Importantly, both the rate and total amount of the accumulated product were strongly affected for the β-globin 5′UTR mRNA by a4Aa20 RNA, with near-complete abolishment of luciferase synthesis at the highest a4Aa20 concentration (Figure 2b).

Collectively, these results show that different modes of eIF4A inhibition can give distinct effects on the translation of mRNAs with different 5′UTRs. The a4Aa20 aptamer has specific translation inhibitory effects depending on the composition (and eIF4A-dependence) of the 5′UTR, indicative of inhibition of eIF4A ‘away from mRNA’. By contrast, AMP-PNP induces a broad, 5′UTR sequence-independent inhibition (compare Figure 1a vs. Figure 2a), suggestive of a different, noncompetitive mechanism ‘on mRNA’, such as that AMP-PNP-bound ATP-dependent RNA-binding protein might attach to 5′UTRs and block SSU scanning.

### 2.3. Presence of AMP-PNP Decreases Efficiency of SSU Complex Assembly at the Start Codon of Capped Poly(A) 5′UTR mRNA

To refine the observed effects and discern if the ‘basic’ scanning machinery could be affected by the ATP-bound form of eIF4A—and at which stage and combination of other factors—we monitored initiation complex assembly on capped poly(A) 5′UTR mRNA in the presence of AMP-PNP using a translation initiation system reconstituted from purified native and recombinant mammalian components. The poly(A) 5′UTR mRNA is not dependent on active scanning, allowing us to observe inhibitory effects other than caused by, e.g., complementarity-based structural impediments. SSU:mRNA complex formation was assessed based on inhibition of reverse transcription (toeprinting) [42,74,93,94]; the resultant cDNA fragments were detected by fluorescent labelling and capillary electrophoresis [16,74]. We introduced several modifications to channel initiation through cap-dependent scanning (see Figure 3a for a schematic of the experiment; more details in Materials and Methods). To increase the probability of cap-guided initiation, we used lower SSU concentration and higher mRNA and eIF4F concentrations in all experiments with the capped and some with the uncapped mRNA, compared to the previously reported conditions [16]. Further, we first assembled the system without mRNA and initiator tRNA (Met-tRNA^fMet^), SSUs and ATP. We used pure native prokaryotic tRNA^fMet^, which we Met-aminoacylated. This avoids introducing potentially competing short (possibly, capped) RNA fragments and nonaminoacylated or noninitiator tRNAs, as found if using total eukaryotic native tRNA or synthetic transcribed tRNA_i_ preparations, thus allowing for high start codon complex output. Unless otherwise indicated, we then added the mRNA and Met-tRNA^fMet^ and preincubated the mixtures at 37 °C for 5 min, to allow factor binding to mRNA. Only after this step, we supplemented the system with the SSUs and ATP (the default in earlier work), AMP-PNP or water, and incubated for a further 15 min prior to reverse transcription. To assess the effects of our modified conditions, we first performed the assay with uncapped poly(A) 5′UTR mRNA in the original published conditions [16], as well as with both uncapped and capped mRNA in modified conditions, each in the presence of ATP (Appendix A; note that the heterogeneity of the cDNA 3′ end length is likely due to the poly(A) 5′UTR 5′ end length heterogeneity resulting from the transcription initiation ‘slippage’ effect when synthetising this mRNA, in similarity to the 5′ ends of late poxviral mRNAs [16,75,76,77,78,79,80]). Translation initiation complex assembly was detectable in the presence of a full complement of factors (eIFs 1, 1A, 2, 3, 4A, 4B and 4F). The modified conditions improved the yield of initiation complex toeprints on the uncapped mRNA (at positions +16, +17 and +18 nt downstream from the first nucleotide of the initiation codon, further referred to as +16 nt signal; compare panels a and b), with still better levels seen with capped mRNA (panel c). As expected from prior work [16,74], omission of eIF2 abolished initiation in each case (bottom traces in each panel), while uncapped mRNA was insensitive to omission of eIF4F (panel b, second trace from top).

We then performed a series of toeprinting assays with capped poly(A) 5′UTR mRNA in the modified conditions, either in the presence of ATP, AMP-PNP or by simply adding water (see Figure 3b for an overview of the results, Figure 4 columns a–c for the aligned fluorescence traces and Appendix A for the unprocessed original traces). We tested this with the omission of the eIF4A/B/F factors in several combinations (c.f. columns vs. rows in Figure 3b and Figure 4). With the full complement of factors, initiation was most efficient in the presence of ATP, intermediate in the water control and lowest with AMP-PNP (Figure 4, columns a–c, row *i*; toeprint at around +16 nt). Further, in the presence of ATP (Figure 4, column a), initiation was most strongly affected by omission of eIF4F (Figure 4, row *v* vs. row *i*); omitting eIFs 4A and 4B in the presence of eIF4F, either singly or together, had intermediate effects (Figure 4, rows *ii–iv* vs. row *i*). In the presence of AMP-PNP (Figure 4, column b), initiation was weak, irrespective of factor omissions (Figure 4, rows *ii*–*viii* vs. row *i*), with eIF4B omission somewhat exacerbating the effect when ‘free’ eIF4A was also present in the system (Figure 4, rows *ii* and *vi* vs. the other rows). Without ATP and AMP-PNP addition (‘water’; Figure 4, column c), initiation was enhanced in the absence of eIF4F, either alone or in combination with codepletion of both, eIF4A and eIF4B (Figure 4, rows *v* and *viii* vs. row *i*).

Several interesting conclusions can be made from these data. First, inhibition by AMP-PNP is clearly seen in the reconstituted system, pointing to inhibition of eIF4A activity as the main explanation for the effects observed in the cell lysate experiments (Figure 1). Second, adding a cap to the poly(A) 5′UTR stimulates initiation by cap-binding factors in the presence of ATP (Figure 3b; Figure 4, column a). This presumably cap-directed mode of initiation in the presence of eIF4F and at least eIF4B or ‘free’ 4A is selectively inhibited by AMP-PNP, implying an active role of eIF4A mediated by ATP hydrolysis when eIF4A is complexed in eIF4F (Figure 3b; Figure 4, columns a–c, rows *i–iii*). The intermediate level of initiation seen in the absence of added nucleotide is insensitive to the depletion of eIF4B and/or ‘free’ eIF4A; it is further mildly stimulated by removal of eIF4F, which is suggestive of a relieved interference between (unproductive) cap-dependent loading and ‘free SSU sliding’ (note also the insensitivity of its noncapped counterpart to omission of eIF4F; Appendix A). That both ATP and AMP-PNP have similar, moderately inhibitory effects compared to the water control in the absence of eIF4F (Figure 3b; Figure 4, columns *v–viii*), suggests a degree of nonspecific inhibition. On the other hand, the data with co-depletion of eIFs 4A and 4B, which shows no difference to the water control, suggests that ‘free’ eIFs 4A and 4B by themselves cannot appreciably block initiation in the presence of AMP-PNP (Figure 3b, Figure 4, row *iv*). Overall, these data support the notion that preventing ATP hydrolysis within eIF4A has the potential to specifically stall initiation, either upon mRNA loading into the SSU mRNA-binding cleft or during scanning, and not just by preventing recruitment to the mRNA cap.

### 2.4. Cap-Guided Initiation Results in the Appearance of an mRNA:SSU Toeprint Upstream of the Usual Start Codon Signal

An intriguing observation in the toeprinting data was the appearance of a prominent additional reverse transcription stall at position −8, −7, −6 nt (termed ‘−8 nt’ toeprint) upstream from the usual +16 nt stop. This was only seen in the presence of both, eIF4F and ATP (e.g., Figure 4, column a, rows *i–iii* vs. all other conditions). Critically, neither did noncapped poly(A) 5′UTR mRNA show such a reverse transcription stall (Appendix A), consistent with previous observations [16], nor there were any additional toeprint signals appearing further in the coding regions in any of the cases (Appendix A). As this mRNA has a homopolymer 5′UTR sequence, it is difficult to envisage the −8 nt complex as the result of sequence-selective recognition. Indeed, an equivalent stop is also noticeable in several of the classic toeprinting studies, which used different capped model mRNAs [42,44,96,97,98,99,100,101,102,103,104]. Nevertheless, to verify the context independence of the −8 nt stop within our conditions, we tested a selection of mRNAs with 5′UTRs of different lengths and sequence composition.

In addition to the mRNA with the poly(A) 5′UTR (27 nt long 5′UTR), we used mRNAs with the same Luc ORF, but preceded with either an (A,U)-containing sequence (37 nt long 5′UTR) or a poly(U) sequence (37 nt long 5′UTR), as well as the natural β-globin mRNA purified from rabbit reticulocyte lysate (53 nt long 5′UTR) (Figure 5a). The (A,U) and poly(U) 5′UTRs included the GGGAAAGC nucleotide sequence at their 5′ ends to promote efficient T7 RNA polymerase transcription initiation and were compliant with the canonical mammalian start codon context (UUAC*ACC***AUG***G*). All mRNAs were capped. We assembled the toeprinting reactions as described in Section 2.3 and assessed the output of reactions with or without added ATP. Compared to the poly(A) 5′UTR mRNA, all other mRNAs demonstrated higher ATP dependence, with poly(A,U) 5′UTR mRNA having an intermediate ATP dependence, and the poly(U) 5′UTR mRNA and β-globin mRNA having the strictest requirements (Figure 5b). Importantly, the −8 nt second toeprint signal appeared prominently with all mRNAs, when ATP was present in the system.

Because the formation of the −8 nt complex is not appreciably 5′UTR context-dependent and its position is not dictated by the distance from the 5′ end of mRNA, its localisation is directly or indirectly dictated by the location of the start codon. What could be the nature of the SSU complex that generates this toeprint? We first considered that our use of methionine-aminoacylated bacterial (*Escherichia coli*) initiator tRNA^fMet^ could be a trigger, as nonoptimal initiator tRNA can lead to unstable intermediates of start codon complex assembly [100]. However, bacterial initiator tRNA^fMet^ was added to all of our assays, including those that did not show −8 nt toeprints and most of the prior studies that showed −8 nt toeprints used cognate eukaryotic Met-tRNA_i_^Met^ [42,44,96,97,98,99,100,101,102,103,104]. Furthermore, we replaced the bacterial Met-tRNA^fMet^ with Met-tRNA_i_^Met^ (as a mixture with total nonaminoacylated *Saccharomyces cerevisiae* tRNAs). This incurred lower overall initiation efficiency but the −8 nt signal was still discernible (data not shown). A more physiological explanation might be in the known conformational change of SSUs from an open, scanning-competent form to a closed form following AUG recognition [66,105] (reviewed in [24]), generating a comparable length of extra 3′-ward protection in footprinting experiments [29]. In this scenario, cap-dependent initiation would somehow slow down interconversion between different intermediates at the start codon (given the −8 nt signal’s dependence on the presence of a cap structure, ATP and eIF4F with eIF4A/4B). This could either function through selective factor recruitment to SSUs at the cap, which are then ‘sent off’ down the 5′UTR (‘cap-severed’ initiation), or more directly through ongoing interactions between the cap and SSUs at start codons (‘cap-tethered’ initiation). Finally, a scenario whereby a second SSU queues upstream of the start codon might also explain the −8 nt stop (see more in Discussion).

### 2.5. Cycloheximide Elongation Arrest Leads to the Accumulation of Multiple SSUs on the Long, Cap-and-Scanning Initiated LL1 5′UTR in the Presence of High Concentration of Magnesium Ions and Nonhydrolysable ATP Analogue AMP-PNP

Given the divergent scenarios of cap-tethered scanning vs. multiple SSU loading invoked above, we sought to use electron microscopy to visualise translation initiation complexes that form under conditions that are exquisitely supportive of the cap-dependent SSU loading onto mRNAs. We also realised that AMP-PNP might exhibit a specific stalling effect on the scanning complexes in an eIF4F-dependent manner, providing additional opportunities to stabilize scanning SSUs as they accumulate along 5′UTRs. Thus, we assembled translation reactions based on mouse Krebs-2 cell lysate, generally similar to those in Section 2.1 and Section 2.2, and programmed them with a Luc-encoding mRNA with LL1 5′UTR containing the 913 nt 5′UTR of the human LINE-1 retrotransposon. mRNAs with this 5′UTR have previously been demonstrated to be translated cap-dependently and scanned efficiently; and Krebs-2 lysates were shown to allow synthetic mRNAs to remain intact for long periods of time [52,106,107,108]. We treated the lysate with micrococcal nuclease immediately before the reaction assembly, followed by a preincubation without mRNA at 30 °C for 5 min. We then added capped LL1 5′UTR mRNA to 30 nM, together with cycloheximide to 1 µg/µL [74], to efficiently stall ribosomes at start codons [109,110], and further incubated the reaction mixtures at 30 °C for 30 min. Previously, in certain conditions, the addition of cycloheximide was shown to stimulate the appearance of ‘halfmers’ (ribosome:mRNA complexes with sedimentation properties marginal between polysomes with natural ribosomal count), which may represent queued-up initiating SSUs [111]. Similar observations were made in cell lysates derived from rapidly formaldehyde-fixed cells where translation was highly efficient [112], or in complexes derived from formaldehyde-stabilised efficiently translating cell-free systems [29,113], with sedimentation, electron microscopy and ribosome footprinting techniques. We thus had a reasonable expectation that due to the known cycloheximide effects in efficiently stalling translation elongation but not cap- and scanning-dependent translation initiation [24,29,74,109,114,115,116,117], we could visualise one or many SSUs nearby a complete ribosome stalled at or nearby start site with cycloheximide. The long and cap- and powered scanning-dependent LL1 5′UTR in this case would facilitate discrimination of such complexes from complexes resulting from cap- or scanning-independent attachment. To help preserve scanning intermediates in their native form, the reaction mixtures were then supplemented with 5 mM AMP-PNP and an additional 10 mM magnesium acetate, similar to our toeprinting reaction conditions (see more in Materials and Methods). Control reactions were assembled with a noncapped version of the same mRNA and omitted the AMP-PNP and magnesium acetate addition, to control for hypothetical unspecific cycloheximide effects on attachment or aggregation of SSUs. We next negatively stained the resultant complexes and imaged them using transmittance electron microscopy.

We were able to detect structures containing multiple particles with features that identify them as SSUs in close proximity to a particle identified as a complete ribosome (Figure 6, also in Appendix A with higher contrast). The control reaction did not result in the appearance of such complexes (Appendix A).

Analysing different electron microscopy images, we could often attribute up to six SSUs to an individual complex, accompanied with one proximal ribosome (Figure 6, also in Appendix A). In many cases, one or several SSUs were in close proximity with the ribosome, or we observed multiple SSUs assembled tightly in polysome-like structures (e.g., Figure 6d; bottom panel). Consistent with this interpretation and the use of cycloheximide, we did not readily detect polysome-like structures assembled from complete ribosomes (even disomes were largely absent), suggesting that the accumulation of the SSUs is not a result of partial polysome disassembly in these conditions. Altogether, and with the caveat that the mRNA itself cannot be discerned under the conditions used, these images appear to visualise scanning intermediates and indicate that multiple SSUs can load and scan a sufficiently long mRNA 5′UTR during cap-dependent translation initiation.

## 3. Discussion

In this study, we present additional data that characterise the mammalian cap-dependent scanning process. We find that during active translation, ATP-bound complexes are required not only for the unwinding of mRNA 5′UTR structures, but can also lead to the inhibition of translation, if ATP hydrolysis and cycling are prevented. This implies either nonspecific interference with translation by the ATP-bound complexes tightly attached along the mRNA, or more specific inhibition where the initiation components themselves require ATP hydrolysis and cycling to yield a successful initiation reaction. We further demonstrate that in the pure initiation system with eIF4A as the only ATP-binding initiation factor available, preventing ATP hydrolysis and cycling specifically inhibits cap-primed initiation when eIF4A is present in its complexed form as eIF4F, together with an excess of ‘free’ eIF4A or eIF4B. This rather unexpected finding confirms that eIF4A has a role beyond ‘clearing structure’ in translation initiation. Our results suggest that eIF4A-catalysed ATP hydrolysis and cycling are required for either cap-dependent loading of the SSUs on mRNA or their efficient (likely, directional) translocation from the cap structures downstream the 5′UTRs toward the start codons, or both, as suggested before [66,67]; reviewed in [24]. Using these ATP effects and elevated magnesium ion concentration to stabilise scanning during highly cap-dependent initiation scenario on a sufficiently long 5′UTR, we directly observed multiple scanning SSUs in the electron micrographs, presumably originating from the same mRNA. These observations strongly argue in favour of the cap-severed initiation model (reviewed in [24]), at least for longer 5′UTRs.

An interesting observation in the electron micrographs of ribosomal scanning is the apparent limit of up to about six SSUs in front of the cycloheximide-stalled ribosome assembled on the ~900 nt long 5′UTR. This is much fewer than could be expected from the ‘theoretical limit’ of ~30, based on the ~30 nt long ribosomal mRNA protection length. One plausible explanation might be that our translation initiation system did not allow loading of a larger number of SSUs due to their molar ratio to mRNA. Indeed, we used a higher concentration of mRNA than optimal for output in the in vitro translation system as measured by luciferase activity. However, we wished to increase mRNA concentration to improve probability of finding the scanning complexes on the electron microscopy grids and had a reasonable expectation that since the bulk of translation is inhibited in this system by cycloheximide, the effective availability of the SSUs will be high enough. Another two possible explanations are that SSUs may either be initially tightly packed but then fall off the mRNA randomly during loading on the electron microscopy grids, or that the SSUs have a tendency not to pack tightly due to some steric restrictions that are not easily contrasted in the electron microscopy images. The latter is consistent with up to ~60 nt SSU protection length observed in the experiments with Edeine or up to 75 nt footprints left on mRNA by the formaldehyde-crosslinked scanning SSUs [2,29]. It has been hypothesised that these extended footprints are due to the coassembly of initiation factors and SSUs on mRNA during powered scanning [67]; reviewed in [24].

A signature of efficient cap- and active scanning-dependent initiation observed in our data is the appearance of the additional SSU toeprint ~8 nt upstream of the usual +16 nt toeprint signal. Other studies with the reconstituted initiation system have shown a similar upstream toeprint (described as the ‘+8 nt’ toeprint, as measured by the 3′-ward distance to it from the hypothetically utilised start codon) [42,44,96,97,98,99,100,101,102,103,104], although it was rarely appearing as prominent as demonstrated in this study. It was best observed with native (and capped) β-globin mRNA [42,44,96,97,98,99,100,101,102,103,104]. One explanation that has been suggested for this was the more ‘open’ configuration of the SSU entry channel in the absence (or diminished activities) of factors such as eIF1A and/or DHX29 and in the presence of eIF1 [70,118]. Indeed, RNase footprinting experiments of stalled translation complexes in vivo have demonstrated strikingly comparable differences in the 3′-ward SSU protection at the start codon, presumably reflecting intermediates of start codon recognition [29] (reviewed in [24]). Measured from the first nucleotide in the P-site of the SSU, the 3′-ward protection extension changed from +6 nt in the ‘early’ complexes to +16 nt in the later complexes (stage comparable to the main +16 nt ’48 S’ complexes assembled here, Figure 7a), resulting in the overall length difference of 10 nt [29]. The difference between 8 and 10 nt can be explained by somewhat different structures of yeast and mammalian SSUs, as well as likely differences in the minimal length of approach to the SSUs for the reverse transcriptase and RNase. It might be that the toeprinting reaction in this case represents an equilibrium between the ‘open’ and ‘closed’ SSU entry channel, or there is a slow dynamic of SSUs ‘closing’ over the start codons and the two separate populations of complexes co-exist (Figure 7b). However, one might assume that the irreversibility of the primer extension would allow detection only of the greater extension length. Yet this is not observed and moreover, it does not explain the apparent absence of the −8 nt signal in the reactions assembled in the same conditions (compared to the ones with capped mRNA) but where a noncapped mRNA was used (e.g., Appendix A).

It is noteworthy that the −8 nt signal was absent in noncapped β-globin mRNA compared to the identical reaction conditions with capped β-globin mRNA [42,70] as well as in most cases when noncapped mRNAs were used, including those coding for IRESes and assayed with limited factor sets [42,44,45,48,50,96,97,101,102,104,119,120,121,122]. Most intriguingly, neither presence of eIF1, absence of eIF1A or DHX29 can promote the appearance of the −8 nt signal in noncapped mRNAs [42,70]. Further, elevating magnesium ions concentration to 8 mM 5 min after complex assembly but prior to the commencement of the reverse transcription did not shift the equilibrium between the normal and more extended toeprints in the same system, suggesting there is no slow dynamics of one configuration changing to another [118]. The same magnesium ion concentration fully blocked start codon complex assembly if added ab initio, suggesting it induces SSU configuration incompatible with loading on mRNA [118]. Conversion of the start codon SSU complexes from eIF2-containing to eIF5B-containing and complete ribosomes resulted in the suppression of the −8 nt signal [98], which is likely due to the additional stabilisation of start codon complexes [51]. Absence of any substantially shifted toeprint signal corresponding to the third, longer variant of the 3′-ward SSU protection over start codons (the +24 nt extension) observed in vivo [29] suggests that these features may not directly correspond to each other (although there is an ~1 nt rearrangement of intensities towards shorter toeprint signal upon LSU joining to the eIF5B-containing reactions) [16,74]. Combining these facts, the toeprinting observations may be difficult to explain solely based on the 3′-ward protection changes of the main start codon SSU complex and the explanation would imply strong effects of cap tethering on start codon recognition and the induction of ‘open’ SSU configuration, possibilities lacking experimental evidence.

To explain the acute dependence of the −8 nt toeprinting signal on the cap, presence of the scanning factors and ATP, as well as its sensitivity to the stabilisation of the main start codon complex attachment to mRNA, we suggest a possible extension to the model, whereby the −8 nt signal could also result from the reverse transcriptase block by a queued second SSU (Figure 7c). Stacking (‘queuing’) of a second SSU behind an SSU positioned over the start codon as part of the post-start-codon-recognition initiation complex has been observed before on mRNA [29,68,69]. In this case, the queued SSU would arrive in its scanning configuration, with the shortest 3′-ward protection of mRNA (+6 nt) [29]. In our previous observations we found evidences of such complexes at different levels. In experiments with sedimentation through sucrose gradients, SSUs resulting from the RNase I disassembly of the in vivo formaldehyde-fixed polysomes (derived from ‘translated mRNA’) demonstrated heterogeneous sedimentation properties and possibly included fast-sedimenting complexes not incompatible with dual ‘stacked’ SSUs (e.g., ‘Heavy’ fraction in Figure 5a, bottom plot of [115]). Further, a peak of 5′ ends located approximately −30 nt away from the start codon and belonging to the start codon-associated footprints with longest 3′ footprint end extension (presumably derived from the late start codon recognition complexes) was detected, resulting in the additional 5′-ward protection of ~18 nt from the regular start codon-associated 5′ footprint ends, an addition consistent with the minimal observed 17–19 nt protection length of the scanning SSUs (Figure 4b, bottom plot in [29]). The same, approximately −30 nt, extension of the footprint 5′ ends from the start codon, is observed for efficiently initiated individual mRNAs, in contrast to slower initiated mRNAs where these complexes were less prominent (YFL039C, YAL005C, YGL123W, YLR208W vs. YDL014W and YGR240C mRNAs in Extended Data Figure 7 of [29]). Both the metagene and individual examples suggest that these complexes are appearing when the early stages of initiation (cap attachment, scanning) are performed faster than the later stages (transition to elongation), consistent with the stacked SSU interpretation, and suggesting that the stacked (queued) second SSU appears predominantly with its minimal protection length over (or interaction with) the mRNA. Based on these evidences and our toeprinting results, we propose that upon the initiation of the reverse transcription and addition of excessive magnesium ions (usually, at least 5 mM ‘free’), the entry channel of the queued scanning SSU closes on mRNA, and assisted by the elongating reverse transcriptase, an insufficiently stable start codon SSU complex can be cumulatively destabilised and ejected from mRNA (Figure 7c). As the 3′-ward protection length of the queued SSU increases by 10 nt during this process, it results in the partial shift of the toeprinting signal 5′-ward during this ‘read through’ event, but for a distance that is smaller than could be expected from the full SSU footprint protection size (Figure 7c). The advantage of this model is that it would completely satisfy the requirement for the cap dependence and the presence of active scanning components (eIFs 4F and 4A/4B plus ATP) to result in efficient SSU stacking, the interplay between SSU start codon complex stabilisation/destabilisation by different eIFs and the irreversibility of the primer extension as not all mRNAs may have the second stacked SSU. The disadvantage is in the toeprint length differences being less consistent with the anticipated SSU protection on mRNA. Regardless of the interpretation, it appears that the occurrence of the −8 nt toeprinting signal is a strong indication of strict cap-dependent, powered-scanning initiation conditions in a pure system with limited factor set.

Overall, using a specific ratio of translation components, capped mRNAs and relatively high magnesium ions concentration, sometimes together with postassembly addition of nonhydrolysable ATP analogue, we developed an approach which leads to the preservation of at least some of the native scanning SSUs, as they remain attached to the 5′UTRs en route to the start codons. Relatively high output of translation or translation initiation complexes in these conditions may promise that the scanning SSUs observed in our systems were a naturally occurring phenomenon, reflecting the mainstream translation route of the live cells. We further detected the scanning SSUs with two different methods, directly visualising with transmission electron microscopy polysome-like structures which can represent ‘ball-on-a-string’ of SSUs assembled on mRNA with long, cap-dependent 5′UTR, or revealing a toeprinting (reverse transcription inhibition) signal from either the start-codon-associated SSUs with the scanning entry channel configuration, or scanning SSUs stacked (immediately adjacent to) the start codon recognition SSU complexes, on different capped, as opposed to uncapped, mRNAs. Importantly, these findings recapitulate conclusions drawn on the metagene level for the footprint (nuclease protection) distribution of the in vivo fixation-stabilised yeast translation complexes, where a substantial difference in 3′-ward nuclease protection of the start codon-associated SSU complexes was attributed to the ‘open’ and ‘closed’ SSU entry channel, and signal characteristic of the second SSU stacked before start codon complex was also evident [29]. The presence of several SSUs in the 5′UTRs, as visualised by electron microscopy, strongly suggests cap-severed translation initiation, at least for long 5′UTRs, and confirms that unless a 5′UTR bears elements impeding scanning, length of the 5′UTR per se is not a limiting factor in translation as soon as it can accommodate multiple SSUs. Our findings also confirm critically important role of both, the complete eIF4F and ‘free’ eIF4A/4B, for the efficient powered scanning and possibly, reconfiguring SSUs to attain more ‘open’ entry channel configuration. It is noteworthy that none of the longer toeprints were observed were any of these factors dropped from the reaction mixtures. Our results suggest that eIF4A (in complex with eIFs 4E, 4B, 4G and SSUs) might function not only as RNA helicase, but also as a protein conferring alternating high/low affinity of the scanning SSU complexes to mRNA, as has been proposed to explain the powered SSU 5′ to 3′ movement [67].

## 4. Materials and Methods

### 4.1. Construction of Plasmids for Run-Off Transcription of Anti-eIF4A Aptamer, Its Scrambled Control RNA and Poly(U)-Luc, Poly(A,U)-Luc mRNAs

Double-stranded DNA fragment with the plus strand sequence 5′TATTATGTC***AAGCTT*CTCTAATACGACTCACTATA**GGGAGACAAGAATAAAACGCTCAAGGGGACCGCGCCCCACATGTGAGTGAGGCCGAAACGTAGATTCGACAGGAGGCTCACAACAGGC***AGATCT***TATTATGTC3′ was constructed by annealing synthetic single-stranded oligonucleotides (Syntol, Moscow, Russia) 5′TATTATGTCAAGCTTCTCTAATACGACTCACTATAGGGAGACAAGAATAAAACGCTCAAGGGGACCGCGCCCCACATGTGAGTGAGGCCGAAACGTAGA3′ and 5′GACATAATAAGATCTGCCTGTTGTGAGCCTCCTGTCGAATCTACGTTTCGGCCTCACTCACATGTGGGGCGCGGTCCCC3′ and extending the strands with *Pfu* DNA polymerase (Promega, Madison, WI, USA), using reaction conditions recommended by the manufacturer, to result in the full duplex DNA. The resulting double-stranded DNA contained (plus strand) ***AAGCTT***
*Hin*dIII and ***AGATCT***
*Bgl*II cleavage sites (highlighted in bold italics), the bacteriophage T7 RNA polymerase promoter sequence (highlighted in bold) and the full sequence of the anti-eIF4A RNA aptamer 20 (a4Aa20; underlined). The resulting double-stranded DNA was digested with the restriction endonucleases and cloned into pObeLucTMV plasmid [74] between *Hin*dIII and *Bgl*II sites to result in pa4Aa20. Cloning results were confirmed by sequencing. For the control scrambled a4AN RNA, same approach was used but the annealed oligonucleotides were: 5′TATTATGTCAAGCTTCTCTAATACGACTCACTATAGGGAGACAAGAATAAAACGCTCAAATGGAAGACGCCAAAAACATAAAGAAAGGCCCGGCGCCAT3′ and 5′CACATAATAAGATCTGCCTGTTGTGAGCCTCCTGTCGAAATGGCGCCGGGCCTTTCTTTATGTTTTTGGCGTCTTCCAT3′, yielding double-stranded DNA fragment with the plus strand sequence 5′TATTATGTC***AAGCTT*CTCTAATACGACTCACTATA**GGGAGACAAGAATAAAACGCTCAAATGGAAGACGCCAAAAACATAAAGAAAGGCCCGGCGCCATTTCGACAGGAGGCTCACAACAGGC***AGATCT***TATTATGTC3′ upon extension.

The same approach as above, but with the complete, 5′-end phosphorylated DNA duplexes (obtained by annealing 5′P-5′AGCTTTTTTTTTTTTTTTTTTTTTTTTTTAC3′ and 5′P-5′CATGGTAAAAAAAAAAAAAAAAAAAAAAAAA3′; 5′P-5′AGCTTATTACAATTACTATTTACAATTACAC3′ and 5′P-5′CATGGTGTAATTGTAAATAGTAATTGTAATA3′) and without the extension/digestion stage was used to construct pTZ10M6Luc and pTZ10M7Luc plasmids, respectively, based on pTZ10ΩLuc [52] with the 5′UTR preceding Luc cut-out between ***AAGCTT***
*Hin*dIII and ***CCATGG***
*Nco*I sites and replaced with the annealed duplexes.

### 4.2. In Vitro Synthesis of RNA

RNAs were synthesised using bacteriophage T7 run-off transcription from linearized plasmid DNA, generally as previously described [74]. For uncapped mRNAs, only all four NTPs were added into the reaction mixtures. For the capped versions, mRNAs were first cotranscriptionally capped by adding 20-fold excess of the antireverse cap analogue (ARCA; NEB, Ipswich, MA, USA or Ambion, Austin, TX, USA) over GTP in the transcription reaction mixtures. pa4Aa20 was digested with *Bgl*II to generate a4Aa20 RNA; pTZ19RbetaLuc was digested with *Sac*I to generate full-length Luc-encoding mRNA (containing 5′UTR with unspecific 42 nt sequence derived from *Boechera divaricarpa* followed by 48 nt of *Xaenopus laevis* β-globin mRNA 5′UTR; overall 5′UTR sequence GGGAAAGCUUUAUUUUUACAACAAUUACCAACAACAACAAACAACAAACAACAUUACAAUUACUAUUUACAAUUACAGUCGACC); pL913Fluc [52] was digested with *Hin*dIII to generate full-length Luc-encoding LL1 mRNA (containing first 913 nt of the human LINE-1 cDNA); pTZA25Luc [16] was digested with *Eco*RI to generate truncated poly(A)-Luc mRNA for toeprinting and with *Sac*I for the full-length Luc-encoding mRNA with poly(A) 5′UTR; pTZ10M6Luc and pTZ10M7Luc were digested with *Eco*RI, to generate truncated poly(U)-Luc and poly(A,U)-Luc mRNAs, respectively. mRNAs were purified as described previously [16,74]. Cotranscriptionally capped mRNAs were next post-transcriptionally capped with ScriptCap m7G Capping System (Epicentre Biotechnologies, Madison, WI, USA) according to the manufacturer’s recommendations, purified again and their integrity was confirmed by denaturing PAGE.

The natural β-globin mRNA was purified from rabbit reticulocyte lysate as described previously, based on salt-induced oligo(dT) beads binding and low-salt elution [16].

### 4.3. Cell-Free Translation

The cell-free translation system was assembled generally as described in [52], using 50% mouse Krebs-2 ascites cell lysate [106,107,108] and creatine phosphate-based ATP regeneration system. The assembly was performed on ice; the translation system was pre-optimised for the added KCl and Mg(OAc)_2_ concentrations, based on the combined maximum yield and product accumulation speed when translating capped 5′β-globin-Luc-3′TMV mRNA (from TZ19RbetaLuc). The lysate was prepared as described before [106]; briefly, Krebs-2 cells were collected in an isotonic buffer (150 mM NaCl, 35 mM Tris-HCl pH 7.5) and centrifuged at 300× *g* followed by pellet resuspension in the same buffer, washed by repeating resuspension and centrifugation for three times, and resuspended in 1.5 cells volumes of hypotonic buffer (10 mM Tris-HCl pH 7.5, 10 mM KOAc, 1.5 mM Mg(OAc)_2_, 2.5 mM DTT) and incubated in this buffer on ice for 20 min. The cells were then disrupted using a Dounce homogeniser, and the lysate was clarified from cell debris by centrifugation for 20 min at 300× *g*, 4 °C, aliquoted and stored at −80 °C. For in vitro translation reactions with Luc luminescence monitoring, an aliquot of the 100% Krebs-2 mouse ascites cell lysate obtained this way was de-frosted immediately before translation reaction assembly and supplemented with 20 mM HEPES-KOH pH 7.6 (at 25 °C), 40 mM KCl, 0.16 mM Mg(OAc)_2_, 2 mM DTT, 2 mM β-mercaptoethanol, 0.5 mM spermidine, 0.2 mM GTP, 16 mM creatine phosphate, 8 mM cAMP, 0.1 mM luciferin, 0.1 mM each of the amino acids, and then ATP or AMP-PNP equimolarly premixed with Mg(OAc)_2_ to desired concentrations. The mixture was further supplemented with 0.1 µg/µL creatine phosphokinase, 1 U/µL RNase inhibitor (RiboLock; Fermentas, Vilnius, Lithuania) and 0.08 µg/µL calf liver tRNA (Novagen, Madison, WI, USA). Upon the assembly, the mRNA amounts needed to obtain the desired final concentrations were quickly mixed in and the reaction mixtures were immediately transferred to 30 °C; their luminescence time course recorded with Chemilum-12 multichannel luminometer (Institute of Cell Biophysics, Pushchino, Russia) at 2.5 s resolution and smoothed with sliding average (Smooth/B = 1/E = 0 100) in Igor Pro (version 6.3.6.4; WaveMetrics, Lake Oswego, OR, USA). For reactions with eIF4A targeting with the aptamer (and the corresponding controls), a4Aa20, a4AN RNAs and *Escherichia coli* 5S rRNA (Boehringer Mannheim, Mannheim, Germany) were mixed in to the desired concentrations, and the reaction mixtures were first incubated for 5 min at 30 °C, prior to the addition of mRNA.

### 4.4. Toeprinting in a Reconstituted Translation System

Toeprinting was performed generally as described before [16,74], with native SSUs, eIF2, eIF4F, eIF3 and β-globin mRNA purified from rabbit reticulocyte lysate (Green Hectares, Oregon, WI, USA), and recombinant *Escherichia coli* Met-tRNA synthetase and recombinant human eIFs 1, 1A, 4A and 4B purified from *Escherichia coli* Z85 transformed with the corresponding overexpressed plasmid vectors, as it was described previously [16,74]. Recombinant truncated poly(A)-Luc, poly(U)-Luc, poly(A,U)-Luc and the full-size native β-globin mRNAs were purified as described in the ‘In vitro synthesis of RNA’ section. Commercially purified *Escherichia coli* MRE 600 initiator tRNA^fMet^ (Sigma-Aldrich, Saint Louis, MO, USA) and total *Saccharomyces cerevisiae* tRNA (Sigma-Aldrich, Saint Louis, MO, USA; size-selected as described previously for total calf liver tRNA [16,74]) were Met-aminoacylated via *Escherichia coli* Met-tRNA synthetase (but not formylated) as described before [16,74].

An increased concentration of mRNA, eIF2 and eIF4F, and decreased concentration of SSUs, compared to the previously described ratios [16,74], were used to ensure high likelihood of cap- and scanning-dependent initiation; the reaction procedure was also modified to include cap-binding step with no added SSUs or ATP. Briefly, 300 nM eIF2, 100 nM eIF3, 150 nM eIF3:eIF4F complex, 750 nM eIF1, 750 nM eIF1A, 250 nM eIF4A and 250 nM eIF4B were mixed on ice in a buffer containing 40 mM Tris–OAc pH 7.5, 1.7 mM Mg(OAc)_2_, 2 mM DTT, 0.25 mM spermidine, 0.4 mM guanosine 5′-[β,γ-imido]triphosphate (GMP-PNP), 0.1 mM EDTA, 110 mM KCl and 0.3 U/μL RiboLock RNAse inhibitor (Fermentas, Vilnius, Lithuania). After that, 40 nM mRNA, 1.6 μM of pure *Escherichia coli* tRNA^fMet^ or 10 μM of total *Saccharomyces cerevisiae* tRNA, both Met-tRNA aminoacylated (as indicated) were added and the mixtures were preincubated at 37 °C for 5 min. Upon preincubation, 50 nM SSUs, 2 mM ATP:Mg(OAc)_2_ or AMP-PNP:Mg(OAc)_2_ (as indicated) were supplemented. The mixtures of final volume of 20 µL were further incubated at 37 °C for 15 min. The reverse transcription, signal acquisition and data analysis were performed exactly as described before [16,74], with reverse transcription initiated with the 5′ [6-carboxyfluorescein](FAM)-GGACTCGAAGAACCTCTG3′ for rabbit β-globin mRNA and 5′ [6-carboxyfluorescein](FAM)-GATGTTCACCTCGATATG3′ for Luc-encoding mRNAs (Syntol, Moscow, Russia). To provide a quantitative estimate of the translation initiation efficiency, we measured fluorescence of the start codon-associated toeprinting peaks (areas under the corresponding region of the curves) in percent of the total 5′UTR fluorescence measured from 5′ end of the 5′UTR to the 3′ end of the start codon-associated region.

### 4.5. Electron Microscopy

Electron microscopy was performed generally as described before [113]. To prepare the samples, 100% Krebs-2 cell lysate (see ‘Cell-free translation’ section) immediately upon defrosting was supplemented with 0.2 U/µL micrococcal nuclease (Fermentas, Vilnius, Lithuania) and 1 mM CaCl_2_ and incubated at 23 °C for 5 min. These conditions were established to achieve near-complete abolishment of translation from mRNAs present in the lysate, without strong inhibition of the added mRNA translation. Immediately, the reaction mixtures were transferred on ice, supplemented with 2 mM EGTA, 40 mM Tris–OAc pH 7.5, 100 mM KCl 2.4 mM Mg(OAc)_2_, 2.5 mM ATP, 0.2 mM GTP, 0.2 mM spermidine, 0.08 µg/µL calf liver tRNA (Novagen, Madison, WI, USA), 0.25 mAU_280nm_/mL purified *Escherichia coli* Met-tRNA synthetase, 0.1 mM each of the amino acids, 1 U/µL RNase inhibitor (RiboLock; Fermentas, Vilnius, Lithuania) and incubated at 30 °C for 5 min. The reaction mixtures were then supplemented with 1 µg/µL cycloheximide and 30 nM capped full-length LL1-Luc mRNA, and incubated at 30 °C further for 30 min. The mixtures were transferred on ice, supplemented with 5 mM AMP-PNP:Mg(OAc)_2_ and 5 mM Mg(OAc)_2_, and gel-filtered via Illustra MicroSpin S300 columns (GE Healthcare, Jefferson City, MO, USA), pre-equilibrated with buffer 40 mM Tris–OAc pH 7.5, 100 mM KCl, 7.4 mM Mg(OAc)_2_, 5 mM AMP-PNP:Mg(OAc)_2_, 0.2 mM GTP, 0.2 mM spermidine. 1 U/µL RNase inhibitor (RiboLock; Fermentas, Vilnius, Lithuania) was added immediately to the gel-filtered mixtures and the solutions were loaded onto carbon-coated grids with surface tension spreading technique (using 40 mM Tris–OAc pH 7.5, 100 mM KCl 7.4 mM Mg(OAc)_2_ buffer for spreading), and contrasted with 1% uranyl acetate solution in water. Imaging was performed via JEM-100C electron microscope (JEOL, Akishima, Japan) with accelerating voltage set to 80 kV.

## Figures and Tables

**Figure 1 ijms-20-04464-f001:**
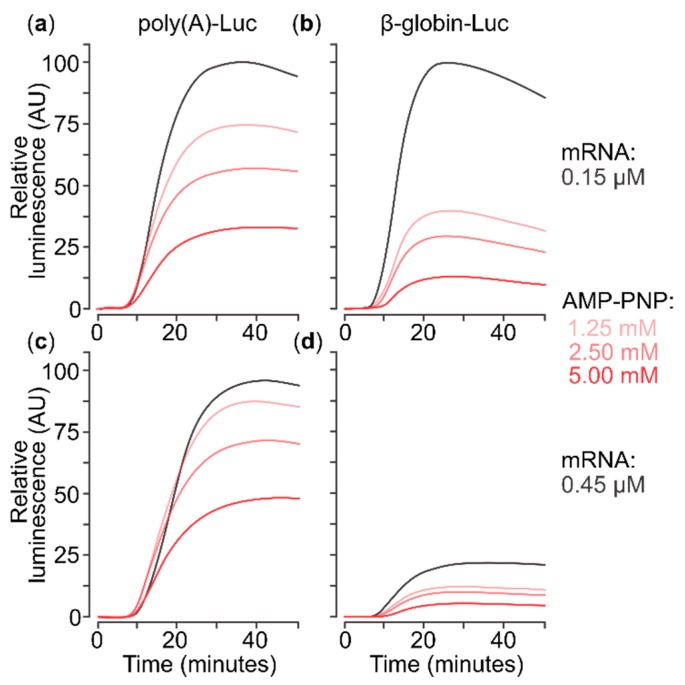
In situ luminescence catalysed by firefly luciferase accumulating during translation of capped mRNAs in a cell-free translation system based on mouse Krebs-2 cells lysate, in the absence or presence of different concentrations of nonhydrolysable ATP analogue (AMP-PNP). (**a**,**c**) Translation of capped 5′poly(A)-Luc-3′TMV mRNA at 0.15 µM (**a**) or 0.45 µM (**b**). (**b**,**d**) Translation of capped 5′β-globin-Luc-3′TMV mRNA. AMP-PNP was added to 1.25, 2.5 or 5 mM (lines colour-coded by shades of red) or omitted (black line).

**Figure 2 ijms-20-04464-f002:**
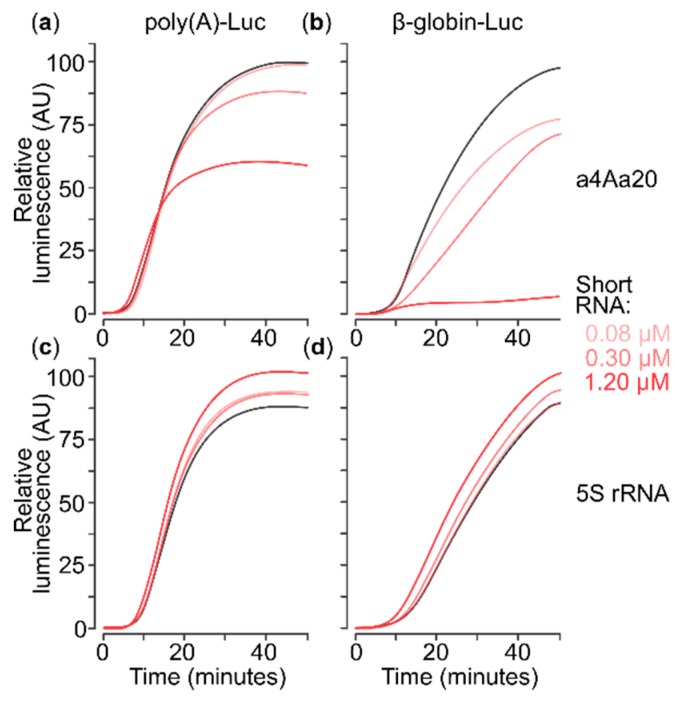
In situ luminescence catalysed by firefly luciferase accumulating during translation of capped mRNAs in a cell-free translation system based on mouse Krebs-2 cells lysate, either without or in the presence of different concentrations of eIF4A-blocking RNA aptamer (a4Aa20). (**a**,**c**) Translation of capped 5′poly(A)-Luc-3′TMV mRNA at 0.15 µM. (**b**,**d**) Translation of capped 5′β-globin-Luc-3′TMV mRNA at 0.15 µM. (**a**,**b**) Addition of eIF4A-blocking RNA aptamer a4Aa20. (**c**,**d**) Addition of *Escherichia coli* 5S ribosomal RNA (rRNA) used as negative control. The short RNAs were added at 0.08, 0.3 or 1.2 µM (lines colour-coded by shades of red) or omitted (black line).

**Figure 3 ijms-20-04464-f003:**
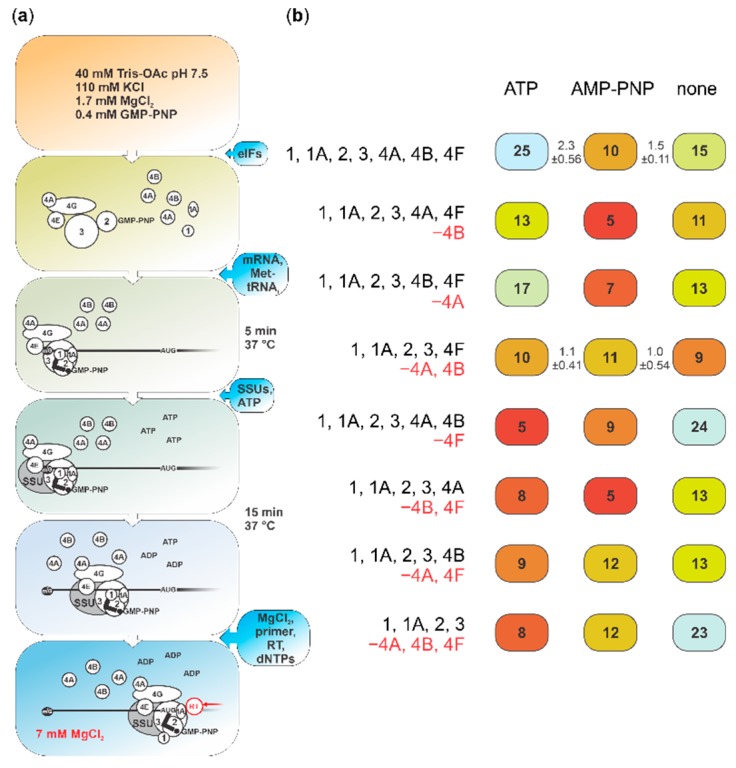
Summary of the analysis of stepwise, cap-guided assembly of ribosome:poly(A)-Luc (capped) mRNA complexes, in the presence or absence of ATP or its nonhydrolysable analogue (AMP-PNP), and different sets of group 4 initiation factors (eIFs). (**a**) Schematic representing steps taken to channel translation initiation predominantly through cap-dependent attachment to mRNA on a cap- and powered scanning-independent mRNA with poly(A) 5′UTR. (**b**) Overview of the results for (a) with the sets of eIFs indicated on left (omitted eIFs shown in red with the minus ‘−’ sign). Numbers in boxes indicate percent of the complex assembly by the fluorescence in the start codon-corresponding toeprint stop relative to the total of the 5′UTR (see Materials and Methods and reference [74] for more detail) and are colour-coded by the complex yield (blue, more; red, less). Numbers between boxes represent mean of complex yield fold change between ATP (left) or no nucleotides (‘none’, right) and AMP-PNP (centre) conditions (using repeat shown in Appendix A), ± double standard deviation.

**Figure 4 ijms-20-04464-f004:**
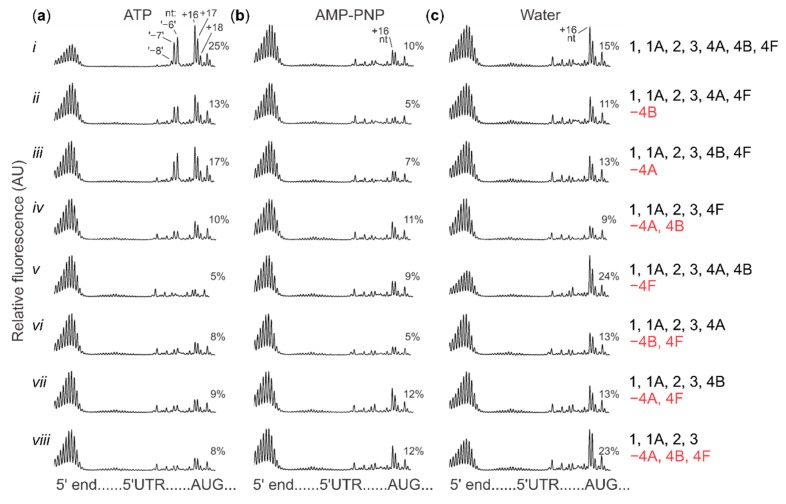
Relative fluorescence of cDNA fragments generated in a reverse transcription reaction with ribosome:poly(A)-Luc (capped) mRNA complexes after electrophoretic separation (toeprint assay). Percent values indicate amounts of fluorescence (area under the curve) corresponding to all signals related to the cognate start codon of this mRNA (+16, +17, +18 nt peaks), relative to the total signal in the 5′UTR. +16 nt denotes the toeprinting peak located 16 nt downstream of the first nucleotide of the start codon, which is considered as the first nucleotide located in the P-site of the SSU:mRNA complex upon start codon recognition. ‘−8 nt’ denotes position of a more 5′-proximal polymerase stop located 8 nt upstream of the +16 nt peak. Capped poly(A)-Luc mRNA at 40 nM was preincubated with mixtures containing different sets of eIFs (*i–viii*; as described to the right of the plots) and *Escherichia coli* Met-tRNA^fMet^ for 5 min at 37 °C. Subsequently, ATP to 2 mM (**a**), AMP-PNP to 2 mM (**b**) or water (**c**), as well as SSUs (all) were added and toeprinting performed as previously described, see Methods for further details and Figure 3 for the experiment schematic and summary of the results. Where ATP or AMP-PNP were added, we assumed 1:1 magnesium ion binding to the solubilized compounds [95] and supplemented the nucleotide triphosphates together with an equimolar amount of magnesium ions using freshly prepared equimolar premixes with magnesium acetate. See Appendix A for replicate assays corresponding to data shown here in rows *i* and *iv*. For the complete unprocessed fluorescence cDNA traces, see Appendix A; note the cDNA signal will appear flipped by the horizontal axis in the unprocessed plots.

**Figure 5 ijms-20-04464-f005:**
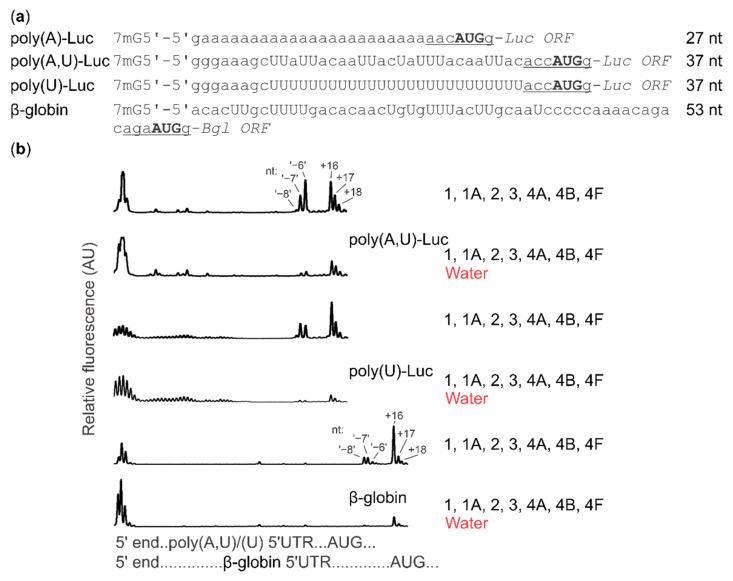
(**a**) Sequences of the synthetic capped mRNAs used in the reconstituted translation initiation system assembly. (**b**) Toeprint assay of ribosome:mRNA complexes assembled on synthetic and natural capped mRNAs. (top two plots), poly(A,U)-Luc mRNA; (middle two plots), poly(U)-Luc mRNA; (bottom two plots), β-globin mRNA. Reactions were assembled as described in Materials and Methods and in Figure 4 legend; reaction mixtures included 40 nM capped mRNAs and were supplemented with either 2 mM ATP or water instead (indicated in the legend on right).

**Figure 6 ijms-20-04464-f006:**
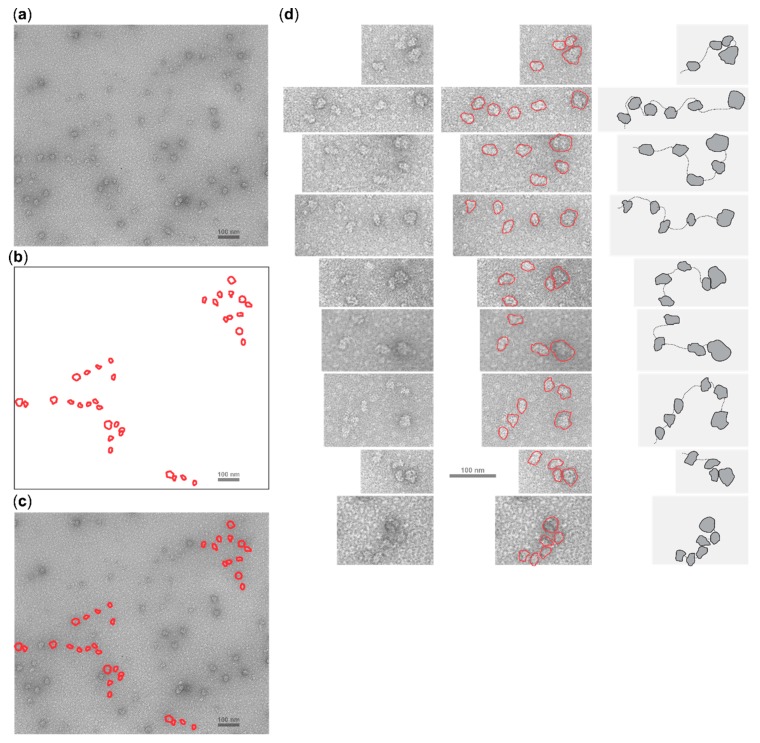
Electron micrographs of, and their possible interpretation for, mRNA:ribosomal complexes assembled in Krebs cells lysate using capped mRNA with LL1 5′UTR. The capped mRNA was preincubated with micrococcal nuclease-treated 50% Krebs-2 cells lysate in the presence of cycloheximide at 30 °C for 30 min and supplemented with 5 mM AMP-PNP and 10 mM magnesium acetate premix. The resultant reaction mixtures were gel-filtered using Illustra MicroSpin S300 columns, contrasted with uranyl acetate and imaged (see more details in Materials and Methods). (**a**) A representative electron microscopy field with particles of ribosome and SSU size and appearance. (**b**) Outlines of rows of SSU-appearing particles (smaller) located close to the singular full ribosome-appearing particles (larger). (**c**) Overlay of (**b**) over (**a**). (**d**, left) Cut-outs from several representative electron microscopy fields with SSUs appearing in rows located close to the singular full ribosomes. (**d**, middle) Outlines of SSUs and ribosomes are overlaid over the electron micrographs. (**d**, right) Schematic with possible interpretation of the (**d**, left) panels; note that the location of mRNA (represented by the dotted line) cannot be predicted from the electron microscopy imaging used in panels (a,c,d) and is used for illustrative purposes only.

**Figure 7 ijms-20-04464-f007:**
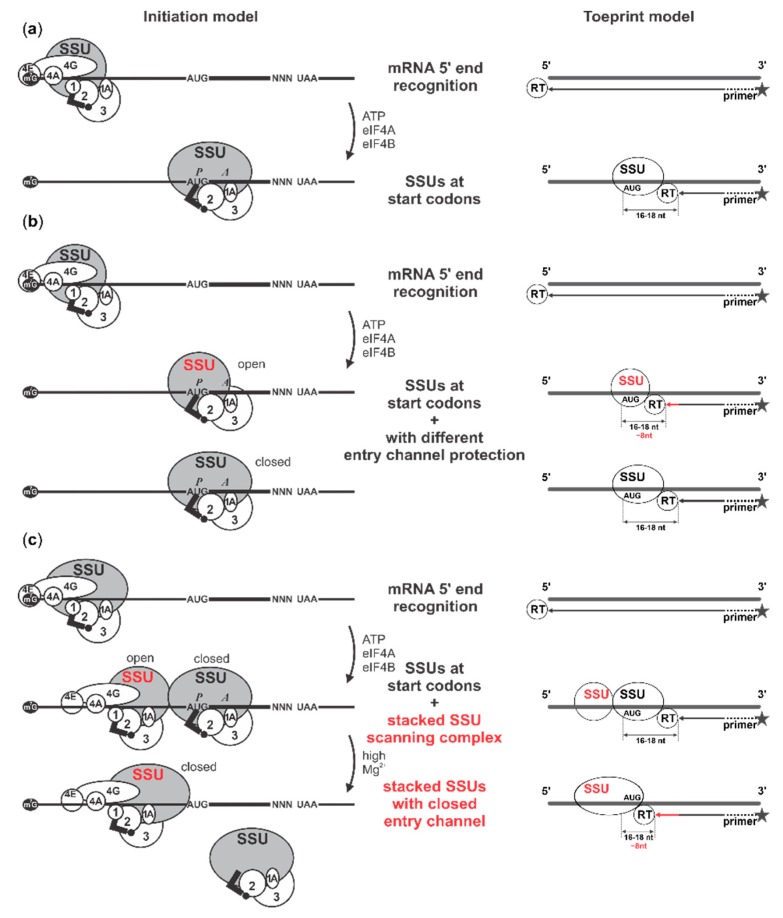
Schematic explaining the detection of ‘open’ entry channel or queuing (stacked) SSUs at the capped mRNAs via toeprinting approach. (left) SSU position over mRNA in the beginning and end of a limiting (**a**,**b**) or excessive (**c**) initiation reaction. (right) Results of the toeprinting reaction for each of the initiation states shown in the left panel. (**b**) The start codon SSU complex in the ‘open’ entry channel configuration stops the reverse transcriptase (RT) at a distance from the usual start codon toeprint signal (right; **b**). (**c**) The stacked SSU scanning complex, together with the reverse transcriptase (RT), displace the start codon SSU complex while adopting the ‘closed’ entry channel configuration, resulting in RT stop at a distance from the usual start codon toeprint signal (right; **c**).

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
