# Peer review of "Migration of Small Ribosomal Subunits on the 5′ Untranslated Regions of Capped Messenger RNA"

_ijms, 2019, doi:10.3390/ijms20184464_

Round 1

Reviewer 1 Report

The manuscript by Shirokikh et al contains a report on the migration of the 40S ribosomal subunit on the 5’ UTR comparing mRNAs with structured and unstructured 5’UTR. As expected, they show that eIF4A is important for scanning of structured, but not unstructured (poly-A) 5’UTR regions. However, AMP-PNP inhibits scanning of both types of mRNAs. Perhaps, the most interesting result is the indication of binding of multiple 40S subunits on the same 5’UTR. These results add details to the understanding of the mechanism of the scanning model. However, the authors should address a number of questions and issues.

Does the binding of multiple 40S subunits to the same 5’UTR depend on the presence of cycloheximide? A control without the drug would be extremely useful, because it has been shown that cycloheximide promotes formation of “halfmers” (Helser TL, Baan RA, Dahlberg AE, Mol Cell Biol. 1981; 1:51-7), although the mechanism of halfmer formation has never been carefully investigated. Furthermore, it has become standard procedure to add cycloheximide to eukaryotic cells before lysis and sucrose gradient centrifugation, because it increases the polysome fraction, but the detailed implications of adding the drug have not been investigated. The experiment reported in Figure 5 of the current manuscript might indicate that artifacts are introduced by the use of cycloheximide. Thus, it could also be informative to see how mRNAs with multiple 40S on the 5’UTR fractionate on a sucrose gradient. “Line 34: ATP-bearing complexes” is slang for the aficionados. Line 167: Why use E.coli 5S rRNA rather than a scrambled aptamer, which would be a better control Lines 209-210: It would be helpful to illustrate this somewhat complicated protocol with a flow diagram as an insert in the figure. Does the E. coli tRNAfmet become formylated? Figure 3 is nice, but it would be useful to show that there are no stops downstream of the initiator region. Statistical analysis of the distribution of stops would also be an important improvement. Lines 256-257: Confusing, because all experiments in Figure 3 are done with capped mRNA. Do the authors mean to say that initiation on the capped polyA UTR is stimulated by eIF4s? Line 261: Typo-midly should be mildly Lines 268-269: An insert with a “score chart” in Figure 3 would be helpful for the non-aficionados.

Author Response

Executive summary

We are very grateful for the thorough review of our manuscript.

All Reviewers provided very positive responses, i.e. ‘the most interesting result is the indication of binding of multiple 40S subunits on the same 5’UTR’ and ‘results add details to the understanding of the mechanism of the scanning model’ (Reviewer #1). ‘These results are of great interest for the translation field since, to my knowledge, such a scanning complex has never been visualised so far. Therefore, I highly recommend this manuscript for publication with minor modifications’ (Reviewer #2). ‘The study adds new knowledge to understand translation initiation and scanning and is of interest to researchers in the mRNA translation field.’ (Reviewer #3).

We acknowledge multiple insightful suggestions made by the Reviewers to improve the manuscript and address them all in the attached point-by-point response.

We provide a revised version of our work. The main changes are: (1) a more thorough discussion of the nature of poly(A) 5'UTR mRNAs and their physiological relevance to late poxviral infection (raised by Reviewers #2 and #3), (2) inclusion of a scrambled RNA control into the description of the anti-eIF4A aptamer effects on translation, (3) inclusion of an experimental flowchart diagram for the toeprinting experiments where capped mRNAs were first pre-incubated with the cap-binding factors, and (4) inclusion of unprocessed fluorescent traces for the main toeprinting results (raised by Reviewer #1). Minor changes and corrections are introduced throughout the text and figures as tracked changes.

We hope that the revised version of our manuscript will now be accepted for publication.

Please see the attachment for the point-by-point response to the reviewer’s comments.

Reviewer 2 Report

In this manuscript, the authors investigated the so-called 5’-3’ scanning mechanism that occurs during translation initiation in higher eukaryotes. For that purpose, they use an in vitro cell-free translation system from mouse Krebs-2 cells and a reconstituted system using pure components containing ribosomal subunits, eIF1, 1A, 2, 3, 4A, 4B and 4F. The translation initiation scanning was monitored by a primer extension assay (also called toe-print assay) with a fluorescent primer. Five models mRNA have been used in this study, luciferase with different 5’UTRs luciferase reporters and the native Beta-globin mRNA that was purified from rabbit reticulocyte lysates. The authors found that a canonical +16 RT arrest that correspond to a pre-initiation complex was efficiently assembled on the AUG start codon. They also detect a -8 RT arrest that is cap-dependent, present only when ATP is added (and not with AMP-PNP) and requires the presence of eIF4A. The authors attributed this -8 toe print to a stalled scanning complex. The -8 toe-print is also detected at the same position with mRNA containing distinct 5’UTRs also in an ATP-dependent manner. In order to confirm that the -8 toe-print is indeed due to a scanning complex, the authors programmed capped luciferase reporter mRNA containing the 913 nt-long LINE-1 5’UTR in a Krebs cell lysate and observed the complexes by Electron Microscopy. They found structures that they interpreted to be an 80S complex next to scanning complexes. These results are of great interest for the translation field since, to my knowledge, such a scanning complex has never been visualised so far. Therefore, I highly recommend this manuscript for publication with minor modifications.  

Major concerns

- In the toe-printing assays, the 5’ end of the signal, which is supposed to correspond the full length RT cDNA is composed of several consecutive bands. This trend is general although in some experiments like in 4(b) the signal looks more homogenous at the 5’ end. Does this mean that there is heterogeneity in the 5’ end of the transcripts that are used or partial capping? The authors should at least comment on this point although this fact does not impact on the presence of the -8 toe-print.

- The Electron micrographs are interesting however, since the mRNA is not visible, the interpretation from the authors is not obvious especially when looking at supplementary figure S5 in which one can see at least one similar structure. As a negative control, I would suggest to repeat this experiment with an mRNA containing a short 5’UTR like globin mRNA for instance. With such an mRNA, multiple scanning complexes, that were observed with the long LINE-1 5’UTR, should be less visible.    

Author Response

Please see the attachment for the complete point-by-point response to the reviewer’s comments.

Executive summary

We are very grateful for the thorough review of our manuscript.

All Reviewers provided very positive responses, i.e. ‘the most interesting result is the indication of binding of multiple 40S subunits on the same 5’UTR’ and ‘results add details to the understanding of the mechanism of the scanning model’ (Reviewer #1). ‘These results are of great interest for the translation field since, to my knowledge, such a scanning complex has never been visualised so far. Therefore, I highly recommend this manuscript for publication with minor modifications’ (Reviewer #2). ‘The study adds new knowledge to understand translation initiation and scanning and is of interest to researchers in the mRNA translation field.’ (Reviewer #3).

We acknowledge multiple insightful suggestions made by the Reviewers to improve the manuscript and address them all in the attached point-by-point response.

We provide a revised version of our work. The main changes are: (1) a more thorough discussion of the nature of poly(A) 5'UTR mRNAs and their physiological relevance to late poxviral infection (raised by Reviewers #2 and #3), (2) inclusion of a scrambled RNA control into the description of the anti-eIF4A aptamer effects on translation, (3) inclusion of an experimental flowchart diagram for the toeprinting experiments where capped mRNAs were first pre-incubated with the cap-binding factors, and (4) inclusion of unprocessed fluorescent traces for the main toeprinting results (raised by Reviewer #1). Minor changes and corrections are introduced throughout the text and figures as tracked changes.

We hope that the revised version of our manuscript will now be accepted for publication.

Reviewer 3 Report

Much remains to be known about the molecular details of eukaryotic mRNA translation initiation and the scanning mechanism after cap recognition. Using a cell-free system, Shirokikh et al. carried out a series of in vitro experiments to examine the consequences of suppression of ATP cycling by eIF4A translation initiation. The authors found that trapping eIF4A in the ATP-bound state inhibited initiation on an eIF4F/A/B-independent mRNA 5'-UTR. They also detected SSU binding around 8 nt upstream of AUG. The authors proposed a model suggesting a cap-severed loading and queuing of multiple SSUs on mRNA. EM imaging of cell-free translation of an mRNA with extended 5'UTR was also suggestive of this model. The study adds new knowledge to understand translation initiation and scanning and is of interest to researchers in the mRNA translation field. Addressing a few points listed below will help improve the manuscript.

More thorough discussion/analysis of evidence from in vivo studies that supports or rejects the model will strengthen the manuscript. Extensive analysis of recent ribosome profiling data from the authors’ group (TCP-Seq) and other groups using various translation initiation inhibitors will help. The authors used mRNA with poly(A)-leader extensively in this study. However (somehow disappointing), the authors didn’t discuss poxvirus post-replicative mRNAs, which have 5’-poly(A) leaders and play an important role in efficiently producing viral proteins. It will improve the physiological relevance of the experimental design by discussing poxvirus mRNAs. Some of the key references are:  

#1. Bertholet, C., E. Van Meir, B. Haggeler-Bordier, and R. Wittek. 1987.VacciniavirusproduceslatemRNAs by discontinuous synthesis.Cell 50:153-162.

#2. Schwer, B., P. Visca, J. C. Vos, and H. G. Stunnenberg. 1987. Discontinuous transcription or RNA processing of vaccinia virus late messengers results in a 5'poly(A) leader. Cell 50:163- 169.

#3. Ahn BY, Moss B. Capped poly(A) leaders of variable lengths at the 5' ends of vaccinia virus late mRNAs. J Virol. 1989 Jan;63(1):226-32.

#4. Yang Z, Martens CA, Bruno DP, Porcella SF, Moss B. Pervasive initiation and 3'-end formation of poxvirus postreplicative RNAs. J Biol Chem. 2012 Sep 7;287(37):31050-60. 

#5. Dhungel P, Cao S, Yang Z. The 5'-poly(A) leader of poxvirus mRNA confers a translational advantage that can be achieved in cells with impaired cap-dependent translation. PLoS Pathog. 2017 Aug 30;13(8):e1006602.

How does the model impact mRNA with shorter poly(A) leader? The studies from aforementioned references (#4 and #5) indicate that most of the poxvirus mRNAs have a poly(A) leader of 8-12 As, although overall their lengths are heterogeneous.

Author Response

(The authors gave the same response as above.)

Round 2

Reviewer 3 Report

The revision improved the manuscript.